# First Report of Endemic Frog Virus 3 (FV3)-like Ranaviruses in the Korean Clawed Salamander (*Onychodactylus koreanus*) in Asia

**DOI:** 10.3390/v16050675

**Published:** 2024-04-25

**Authors:** Jongsun Kim, Haan Woo Sung, Tae Sung Jung, Jaejin Park, Daesik Park

**Affiliations:** 1Division of Science Education, Kangwon National University, Chuncheon 24341, Republic of Korea; jongsun331@naver.com (J.K.); zhqnfth1217@naver.com (J.P.); 2College of Veterinary Medicine, Kangwon National University, Chuncheon 24341, Republic of Korea; sunghw@kangwon.ac.kr; 3College of Veterinary Medicine, Gyeongsang National University, Jinju 52828, Republic of Korea; jungts@gsnu.ac.kr

**Keywords:** endemic, infectious disease, phylogeny, strain, whole genome sequence

## Abstract

*Frog virus 3* (FV3) in the genus *Ranavirus* of the family *Iridoviridae* causes mass mortality in both anurans and urodeles worldwide; however, the phylogenetic origin of FV3-like ranaviruses is not well established. In Asia, three FV3-like ranaviruses have been reported in farmed populations of amphibians and reptiles. Here, we report the first case of endemic FV3-like ranavirus infections in the Korean clawed salamander *Onychodactylus koreanus*, caught in wild mountain streams in the Republic of Korea (ROK), through whole-genome sequencing and phylogenetic analysis. Two isolated FV3-like ranaviruses (*Onychodactylus koreanus* ranavirus, OKRV1 and 2) showed high similarity with the *Rana grylio* virus (RGV, 91.5%) and *Rana nigromaculata* ranavirus (RNRV, 92.2%) but relatively low similarity with the soft-shelled turtle iridovirus (STIV, 84.2%) in open reading frame (ORF) comparisons. OKRV1 and 2 formed a monophyletic clade with previously known Asian FV3-like ranaviruses, a sister group of the New World FV3-like ranavirus clade. Our results suggest that OKRV1 and 2 are FV3-like ranaviruses endemic to the ROK, and RGV and RNRV might also be endemic strains in China, unlike previous speculation. Our data have great implications for the study of the phylogeny and spreading routes of FV3-like ranaviruses and suggest the need for additional detection and analysis of FV3-like ranaviruses in wild populations in Asian countries.

## 1. Introduction

Members of the genus *Ranavirus*, belonging to the family *Iridoviridae*, infect a wide range of ectothermic vertebrates including fish, amphibians, and reptiles in both wild populations and those in commercial farms [1,2,3]. Ranavirus infection is one of the well-known causes of the global decline of the amphibian population [2,4,5,6]. Outbreaks of ranaviruses in amphibians have occurred worldwide, including in the United States of America (USA), Canada, Republic of Korea (ROK), Japan, China, and many countries (such as the United Kingdom (UK), Germany, France, Brazil, and Chile) in Europe and South America [7,8]. The World Organization for Animal Health (OIE) has designated ranaviruses as notifiable diseases in amphibians [9]. Ranaviruses, which mainly infect amphibians, include three groups: *Frog virus 3* (FV3), *Common midwife toad virus* (CMTV), and *Ambystoma tigrinum virus* (ATV) [10,11]. Among them, the FV3-like ranaviruses most widely infect amphibian species [7,8,10,12,13].

FV3-like ranaviruses cause mass mortality in both anurans and urodeles, particularly across many countries in America and Europe. Identifying the phylogeny and origin of the FV3 is likely to reveal evolutionary steps such as changes in the host range and worldwide spreading of the virus [12,14]. Global lineages of FV3-like ranaviruses are largely divided into three groups: the New World clade, which is a mix of North American and European lineages (e.g., FV3; spotted salamander Maine virus, SSME; Frog virus 3 isolate RUK13, RUK13; and Frog virus 3 strain Z994, Z994), the Asian clade (e.g., soft-shelled turtle iridovirus, STIV; *Rana grylio* iridovirus, RGV; and *Rana nigromaculata* ranavirus, RNRV), and the remaining clades (e.g., Bohle iridovirus, BIV; German gecko ranavirus, GGRV; and zoo ranavirus, ZRV). The last mixed clade, which includes FV3-like ranaviruses from countries in different continents (Australia, Germany, and the USA), is debatable. Additionally, whether FV3-like ranaviruses originated either in America or Asia remains debatable. When analyzing global FV3-like ranaviruses, the Asian clade, detected in China, is located at a more basal root of the phylogenetic tree than the New World clade [14,15,16]. However, some North American FV3-like ranaviruses are also located in the same clade, although relevant whole genome data are not available yet [14]. To date, three Asian FV3-like ranaviruses (STIV, RGV, and RNRV) have been reported in cultured or farmed amphibians and reptiles [17,18,19]. The possibility that ranid frogs such as *Rana grylio* and *Lithobates catesbeianus*, which are imported from the USA, transferred FV3-like ranaviruses to farmed or endemic frogs through a spillover process in China has been repeatedly suggested [14,18,20]. Unfortunately, although FV3-like ranaviruses have been reported in wild Asian species [7], none have been verified in wild populations through whole genome sequencing, hindering further studies in this regard.

Here we report the first case of endemic FV3-like ranavirus infections in a wild hynobiid salamander (Korean clawed salamander, *Onychodactylus koreanus*) in Asia, caught from a mountain population in the ROK, using whole-genome sequencing and phylogenetic analysis. We isolated two FV3-like ranaviruses from infected *O. koreanus*, obtained their whole genome sequences, and determined their genomic and phylogenetic characteristics.

## 2. Materials and Methods

### 2.1. Sample Collection and PCR Diagnosis

Two diseased larvae of the Korean clawed salamander (*O. koreanus*) were collected from a mountain stream in Chuncheon-si (City), Republic of Korea (37.840727° N, 127.798621° E) between December 2022 and March 2023. We first described external pathological symptoms and euthanized them by cooling them down in 500 mL of ice and water (5:1, 2–4 °C) for 15–30 min [21] followed by decapitation using a guillotine [22]. *Onychodactylus koreanus* was collected with permission granted by the Mayor of Chuncheon-si (#4180 00085202 200004). All experimental procedures were reviewed and approved by the Institutional Animal Care and Use Committee of the Kangwon National University (KW-221006-2).

To investigate possible ranavirus infection, we dissected and collected liver and kidney samples from each salamander following OIE procedures [23]. Genomic DNA was extracted from a mixture of the liver and kidney using a DNeasy Blood and Tissue Kit (Qiagen, Hilden, Germany) following the protocol described by the manufacturer. Extracted DNA was quantified with the Qubit 1X dsDNA HS Assay Kit (Invitrogen, Waltham, MA, USA) using a Qubit3 Fluorometer (Invitrogen, Waltham, MA, USA) and kept at −80 °C until further processing.

To detect ranavirus infection, we amplified partial sequences of the major capsid protein (MCP) gene of ranaviruses using the primer pair MCP4 and MCP5 [24]. PCR amplification was performed in a SimpliAmp Thermal Cycler (Thermo Fisher Scientific, Waltham, MA, USA) following the methods described in our previous studies [25,26,27]. We included a negative control of molecular biology-grade water (HyClone, Waltham, MA, USA) and a positive control of DNA from *Kaloula borealis*, in which we confirmed the occurrence of a ranavirus infection in 2016 [28]. The amplified PCR target products were confirmed on a 1% agarose gel and sent to Macrogen, Inc. (Seoul, Republic of Korea) for sequencing. We aligned the obtained DNA sequences using Geneious Prime [29] and confirmed the occurrence of a ranavirus infection using the Basic Local Alignment Search Tool (BLAST) at NCBI (https://blast.ncbi.nlm.nih.gov/Blast.cgi (accessed on 15 June 2023)).

### 2.2. Virus Isolation and Cell Culture

To isolate the viruses, we pooled the tissues of both the liver and kidney of each salamander and homogenized the tissues with a mortar and pestle in 2 mL of phosphate-buffered saline (PBS, pH 7.4) containing 10% antibiotics–antimycotics (Gibco, Grand Island, NY, USA). The suspensions were centrifuged at 3000 rpm for 10 min, and supernatants were filtered through a Millex-HV 0.45 µm pore-size syringe filter unit (NovaTech, Kingwood, TX, USA). The isolates were propagated in epithelioma papulosum cyprini (EPC) cells until a cytopathic effect (CPE) was observed.

EPC cells were cultured in 75 cm^2^ flasks (SPL, Seoul, Republic of Korea) at 22 °C with 5% CO_2_ in Dulbecco’s Modified Eagle Medium (DMEM, Gibco, Grand Island, NY, USA) supplemented with 10% fetal bovine serum (FBS, Gibco, Grand Island, NY, USA) and 1% antibiotic–antimycotics, as described previously [30]. EPC cells were sub-cultured (1:4) every 5–6 days by washing with PBS and treated with 3 mL of 0.25% trypsin and 0.53 mM EDTA (Gibco, Grand Island, NY, USA) to detach adherent cells [31]. EPC cells were collected by centrifugation at 1000 rpm for 10 min and resuspended in the growth medium. After inoculation of virus isolates, EPC cells were maintained at 22 °C in Dulbecco’s Modified Eagle Medium (DMEM) supplemented with 2% FBS, and 1% antibiotics–antimycotics. Ranavirus-infected cells showing CPE were stored at −80 °C.

### 2.3. DNA Extraction, Whole Genome Sequencing, Assembly, and Genome Annotation

Genomic DNA was extracted from the infected EPC cells using the Maxwell RSC Viral Total Nucleic Acid Purification Kit (Promega, Madison, WI, USA). A DNA library was prepared using a TruSeq Nano DNA kit (Illumina) and sequenced on a MiSeq platform (Illumina) at Macrogen Inc. We performed de novo assembly of the paired-end reads using SPAdes software v3.5.0 [32], while scaffolding was conducted using Platanus v2.2.2 [33]. We predicted the open reading frames (ORFs) for the genomic sequences of the two isolated FV3-like ranaviruses using GeneMarkS [34] and the Genome Annotation Transfer Utility (GATU) [35] based on the reference FV3 genome (GenBank Accession No. AY548484; [36]). ORFs were identified according to the following criteria, as established in a previous study [37,38]: (1) the sequence length was at least 120 bp, (2) the predicted ORFs were not located within larger ORFs, and (3) overlapping ORFs had homologs in other sequenced ranaviruses. Annotations were selected based on 80% identity matches with a database of all ranavirus CDS sequences published in NCBI (accessed on 15 June 2023). The hypothetical protein annotations, which showed lower-scoring matches were manually checked and judged as to whether they contained functional information.

We collected three amphibian-like ranavirus genomes (ATV, AY150217; CMTV, JQ231222, and FV3) for a detailed comparison of the ORFs between the ranaviruses from this study (OKRV1 and 2) and other ranaviruses using the BLASTP program. Among the FV3-like ranaviruses, we selected six strains of FV3-like ranaviruses based on the phylogenetic tree of FV3-like ranaviruses from previous studies [12,14,16]: three strains were isolated in China (STIV, EU627010; RGV, JQ654586; and RNRV, MG791866), whereas the other strains were isolated in the USA (FV3, AY548484), Canada (Z377, MK959608), and Australia (BIV, KX185156) (Appendix A). We calculated the overall ORF similarity, functional ORF similarity, and hypothetical protein similarity by averaging each of the 104, 45, and 59 individual ORF similarities between OKRV1 and OKRV2 and the six strains of FV3-like ranaviruses (Appendix A).

### 2.4. Phylogenetic Analyses

For phylogenetic analyses, we gathered and aligned 48 ranavirus genomes, including 37 strains of FV3-like ranaviruses, 10 strains of other ranaviruses (CMTV-, ATV-, and Epizootic haematopoietic necrosis (EHNV)-like ranaviruses), and a short-finned eel ranavirus (SERV, KX353311) as an outgroup (Appendix A). We extracted 45 core genes (26 functional and 19 hypothetical protein genes, Appendix A) from each ranavirus genome based on the previous phylogenetic study [14], which used a total of 49 ORFs. We omitted the four ORFs (ORF3R, two ORF56Rs, and ORF58R), which were not applicable in OKRV1 and OKRV2. We concatenated the core genes (43,731 bp) using Geneious Prime (Appendix A). In the previous study [14], ORFs of the 45 core genes showed over 80% amino acid homology in a genus-wide pan-genome analysis. We constructed a maximum likelihood (ML) phylogenetic tree on the IQ-TREE web server (http://iqtree.cibiv.univie.ac.at/ (accessed on 24 March 2024)) [39,40] using 1000 ultra-fast bootstrap replicates. The TIM+F+R2 model was selected under the Bayesian information criterion as the best-fit model using IQTREE’s built-in ModelFinder program according to the Bayesian information criterion [41].

## 3. Results

### 3.1. Field Observation and PCR Diagnosis

In December 2022, we identified the first larva of *O. koreanus* to show signs of ranavirus infection in a mountain stream in Chuncheon-si, Republic of Korea. We identified skin lesions and edema in the trunk, head, and limbs. In March 2023, another diseased larva of *O. koreanus* was identified at the same site. We observed the clinical signs of ecchymosis on the ventral side of the body and edema throughout the body (Figure 1). The two amplified MCP sequences from the larvae showed 100% homology with FV3-like ranaviruses (Z994, MK959621, and SSME, KJ175144).

### 3.2. Virus Isolation, Assembly, and Genome Annotation

Cultures of both FV3-like ranavirus isolates showed typical CPE following ranavirus infection. The CPE began with the rounding of the cells, progressing to cell detachment in the cell monolayer, followed by cell lysis at 3 days after inoculation (Figure 2). The two ranavirus isolates were designated *Onychodactylus koreanus* ranavirus 1 and 2, respectively (OKRV1, PP518040 and OKRV2, PP518041). De novo assembly of the paired-end reads for OKRV1 and OKRV2 produced a contiguous consensus of two sequences of 104,734 and 104,626 bp, with an average coverage of 10,448 and 10,522 reads/nucleotide, respectively, and a G+C content of 55.1%. Both ranavirus genomes encode 104 ORFs (encoding 40 functional and 64 hypothetical proteins) ranging in size from 43 to 1294 amino acids (AA) in length. The ORF similarity between OKRV1 and OKRV2 was 99.8%, and most ORFs (87 of 104) had 100% homologous similarity. The remaining 17 ORFs had homologous similarities ranging between 94.7 and 99.8%.

In BLASTP searches, the overall similarity of ORFs between OKRV1 and other ranaviruses ranged from 92.2% (RNRV) to 69.6% (ATV) (Table 1 and Appendix A). We separately presented the similarity for functional ORFs or hypothetical proteins in Appendix A. RNRV and RGV showed >90% similarity with OKRV1, whereas the remaining viruses, except ATV, showed between 86.2% and 82.6% similarity (Table 1 and Appendix A). The sum of the number of not applicable (BLASTP *E* value > 0.001; NA) ORFs and ORFs that had less than 90% similarity out of 104 ORFs ranged from 10 (9.6% in RNRV) to 34 (32.7% in ATV) (Table 1). We found either the ORFs which show less than 90% similarity or NA in 10 different functional ORFs (15R, 26R, 33R, 35R, 41R, 47 L, 52 L, 54 L, 67R, and 84R). Among them, the 26R (truncated putative e1F-2alpha-like protein), 33R (neurofilament triplet H1-like protein), and 47 L (neurofilament triplet H1 protein CDS) ORFs showed greater variation than the others among the eight compared ranaviruses. 

### 3.3. Phylogenetic Analyses

In the phylogenetic tree constructed using the 45 core ORF genes from whole-genome sequencing of 37 FV3-like ranaviruses, three distinct lineages were identified (Figure 3): the New World clade, the Asian clade, and the remaining clade. The basal position of out-group ranaviruses (CMTV-, ATV-, and EHNV-like ranaviruses) to FV3-like ranaviruses in the three was similar to that observed in previous studies [3,12,14]. OKRV1 and 2 formed an independent clade within the monophyletic clade of Asian FV3-like ranaviruses (STIV, RGV, and RNRV). The Asian clade is a sister group of the New World clade.

## 4. Discussion

In this study, we report the first case of an endemic FV3-like ranavirus infection of the hynobiid salamander (*O. koreanus*), which was found in a wild mountain population in Asia, based on whole-genome analysis. In Asia, three cases of FV3-like ranavirus infections have been reported in China, all of which were found in either cultured or farmed amphibians and reptiles [17,18,19]. OKRV1 and 2, isolated in this study, showed greater (>91%) ORF homology with RGV and RNRV from two amphibian species than other ranaviruses compared in the study, including STIV and FV3-like ranaviruses from North American amphibians. In the phylogenetic tree, OKRV1 and 2 formed an independent clade within the Asian clade, a sister group to the New World clade. Our results are the first report on endemic Asian FV3-like ranaviruses and suggest that we cannot exclude the possibility that FV3-like ranaviruses reported in China might also be local Asian FV3 strains and not imported from North America, as previously suspected.

The isolated OKRV1 and 2 are likely endemic FV3 strains in the Asian clade. Three lines of evidence support this hypothesis. First, we collected infected salamanders from the upper areas of mountain streams, where imported or invasive amphibians and reptiles have never been introduced or exposed. In previous studies, we detected FV3-like ranaviruses in both pristine areas and areas exposed to invasive species, such as *L. catesbeianus* and *Trachemys scripta elegans* in the ROK [25,26,27,42]. In our previous study, ranaviruses were located in different phylogenetic nodes based on habitat sharing with *L. catesbeianus* [27]. Secondly, in our phylogenetic tree, OKRV1 and 2 formed independent clades and clade residues within the known Asian clade. Finally, OKRV1 and 2 showed meaningful differences from all other known FV3-like ranaviruses in ORF comparisons. The closest RNRV (92.2% similarity) showed a 9.6% difference in the combined number of NA ORFs and ORFs with less than 90% homology. In particular, in the comparison of functional ORFs, OKRV1 and 2 and RNRV did not share the truncated putative e1F-2alpha-like protein (26R) and showed 85.94% homology with the neurofilament triplet H1-like protein (33R) ORFs. The latter protein has been repeatedly used in phylogenetic and differentiation studies of ranaviruses [30,43,44]. These results, taken together with those of previous studies, indicate that OKRV1 and 2 are new FV3-like ranaviruses that are endemic to local areas.

Our ORF comparison and phylogenetic results suggest that RGV and RNRV, which were detected in China, may also be endemic Asian strains. In previous studies, RGV and RNRV were considered to be imported from North America through the direct international trade of amphibians, following direct contact, or through spill-over processes to farming animals [18,20]. However, to date, no clear evidence for this has been provided. RGV was isolated from *R. grylio*, which has been cultured in the Quanqiao Experimental Field (Wuhan, China) [45,46]. Although *R. grylio* originally spread globally from North America, breeding populations have been established in many countries [46,47]. Therefore, *R. grylio* could be locally infected by endemic FV3 strains in China. Mu et al. [20] insisted that RNRV, which infects farming frogs, moved into the farm from nearby river water. The farm in the study carried out by Mu et al. is located in Inner China, Sichuan Province [20], suggesting possible infection by local endemic FV3 strains. To further support this suggestion, RGV and RNRV were found to be the closest to OKRV1 and 2 in the ORF comparison. In China, although FV3-like ranavirus infections were recently detected in wild amphibian populations, nobody has phylogenetically studied and investigated the endemic possibility based on whole-genome sequencing [7,48,49]. However, considering our ORF comparison, STIV, which was also detected in China, may not be an endemic Chinese strain, unlike RGV and RNRV. STIV was detected in farmed reptiles and soft-shelled turtles in Shenzhen, a coastal city in China [50]. Therefore, the origin of STIV might be different from either RGV or RNRV. To clarify this, whole-genome sequencing of FV3-like ranaviruses found in wild populations in China is necessary.

FV3-like ranaviruses can infect hynobiid salamanders, a primitive urodele group widely distributed in Asian countries. To date, there are at least three urodele genera (*Ambystoma*, *Notophthalmus*, and *Triturus*) that can be infected by FV3-like ranaviruses [51,52,53,54,55,56]. However, prior to this study, there were no case reports from mainland Asia. Although *Hynobius nebulosus* may have been infected with an FV3-like ranavirus in Japan [57], this was not clearly confirmed based on the whole genome sequence. In addition, the origin of infection, whether endemic or imported, was not verified. In Asian countries, including the ROK, *Onychodactylus* spp. inhabits the upper areas of mountain streams [58], where imported or invasive amphibians and reptiles are rarely introduced or exposed. Thus, the results from our study indicate that FV3-like ranaviruses exist in remote forested areas, as well as in lowland areas in mainland Asia. In previous studies, we detected FV3-like ranaviruses in both pristine and lowland areas [25,26,27]. Our results should be meaningful for preventing wild populations of mountain salamanders from being infected by ranaviruses and for conserving endangered lowland-breeding *Hynobius* spp. in the ROK and other Asian countries.

In order to clearly confirm the existence of endemic FV3 strains in mainland Asia, a few factors need to be further taken into account. First, even though endemic FV3-like ranaviruses have been found in the Chuncheon population of *O. koreanus*, more research is still required in other populations. Second, studies are also required to determine whether endemic FV3 strains are present in other native amphibian species. Third, further investigation is required, encompassing the viruses that have been discovered in North America and tentatively classified into the Asian FV3-like ranavirus lineage [14]. The comparative analyses of genetic characteristics among the viruses that were previously found in China and North America and that were found in this study are necessary. Finally, to clarify the phylogeny of FV3-like ranaviruses, whole-genome studies on ranaviruses in a variety of Asian ectotherms, such as fish and reptiles, should also be initiated.

In this study, we reported the first case of endemic FV3-like ranaviruses based on whole-genome analysis in mainland Asia. This is the first clear case of infection of a hynobiid salamander caused by FV3-like ranaviruses. OKRV1 and 2, isolated in our study, showed closer similarity to RGV and RNRV, which were found in China, than to other FV3-like ranaviruses in ORF comparisons and formed a monophyletic clade in the phylogenetic tree. These results strongly suggest the existence of endemic FV3 strains in mainland Asia. Our results have great implications for the study of the phylogeny and routes of spread of FV3 and suggest an urgent need for further studies of additional detection and analysis of FV3-like ranaviruses in wild populations in Asian countries.

## Figures and Tables

**Figure 1 viruses-16-00675-f001:**
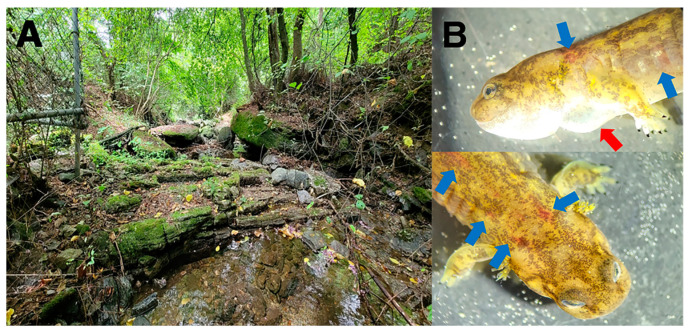
The study site and *Onychodactylus koreanus*. Landscape of the study site (**A**) and a diseased *O. koreanus* ((**B**); red arrows indicate edema and blue arrows indicate ecchymosis), observed at the study site.

**Figure 2 viruses-16-00675-f002:**
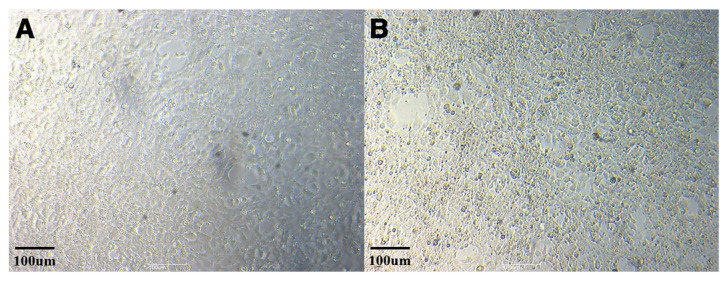
Cytopathic effect (CPE) caused by ranavirus isolate OKRV1 in EPC cells (uninfected normal EPC cells (**A**); infected EPC cells with OKRV1 (**B**)).

**Figure 3 viruses-16-00675-f003:**
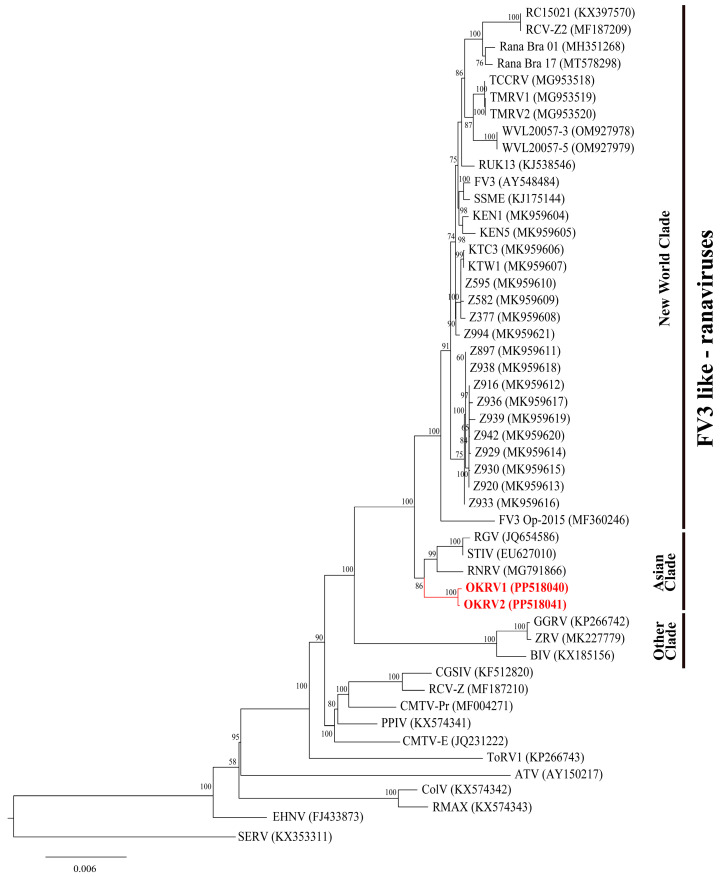
Maximum likelihood phylogenetic tree showing the relationships among OKRV1 and OKRV2 (labeled in red) and 37 FV3-like ranaviruses, 10 other ranaviruses, and short-finned eel ranavirus (SERV, KX353311) as outgroups based on the concatenated 45 core genes (43,731 bp). The bootstrap values are indicated at internodes. The GenBank Accession numbers are indicated in parentheses.

**Table 1 viruses-16-00675-t001:** Characterization of predicted open reading frames (ORFs) of OKRV1 and OKRV2. Only 10 functional ORFs out of a total of 104 ORFs, at least having either not applicable or <90% homology among the eight compared ranaviruses, were shown. The full version of Table 1 is presented in Appendix A. ID, identity; NA not applicable.

OKRV1 (Republic of Korea) (PP518040)			OKRV2 (Republic of Korea) (PP518041)		RGV (China) (JQ654586)		RNRV (China) (MG791866)		STIV (China) (EU627010)		FV3 (United States of America) (AY548484)		CMTV-E (Spain) (JQ231222)		ATV (United States of America) (AY150217)	
ORF /AA	Nucleotide position	Predicted function	ORF /AA	ID (%)	ORF /AA	ID (%)	ORF /AA	ID (%)	ORF /AA	ID (%)	ORF /AA	ID (%)	ORF /AA	ID (%)	ORF /AA	ID (%)
15R /276	17585-18412	Putative integrase-like protein	15R /276	99.6	17R /276	99.3	17R /276	99.3	NA /NA	0.0	16R /276	98.2	93L /276	97.5	NA /NA	0.0
26R /70	31719-31928	Truncated putative e1F-2alpha-like protein	26R /70	100.0	28R /70	100.0	NA /NA	0.0	30R /59	93.0	26R /77	86.9	81L /271	94.4	57R /260	78.7
33R /688	37230-39293	Neurofilament triplet H1-like protein	33R /698	98.3	34R /645	87.5	34R /627	85.9	35R /645	85.9	32R /630	77.5	76L /717	76.9	61R /739	78.6
35R /107	39711-40031	L-protein-like protein	35R /107	100.0	36R /107	99.1	36R /107	99.1	38R /107	99.1	34R /107	100.0	74L /107	95.3	NA /NA	0.0
41R /117	44186-44536	Putative hydrolase of the metallo beta lactamase superfamily	41R /117	100.0	42R /117	99.2	42R /117	99.2	43R /117	99.2	39R /117	97.4	68L /117	93.2	66R /89	87.6
47L /130	51222-51611	Neurofilament triplet H1 protein CDS	47L /130	100.0	47L /145	94.6	47L /141	81.0	49 L/193	69.1	46L /82	88.3	62R /156	76.6	72L /276	62.7
52L /356	56047-57114	Putative 3-beta-hydroxy-delta 5-C27 steroid oxidoreductase-like protein	52L /356	100.0	52L /356	99.7	52L /356	99.7	54L /356	99.7	52L /356	100.0	57R /356	98.3	52R /54	77.1
54L /77	59219-59449	Putative nuclear calmodulin-binding protein	54L /77	100.0	54L /77	98.7	54L /77	100.0	NA /NA	0.0	54L /77	100.0	NA /NA	0.0	NA /NA	0.0
67R /96	74055-74342	Putative interleukin-1 beta convertase precursor	67R /96	100.0	68R /96	95.8	68R /96	95.8	67R /96	95.8	64R /96	99.0	43L /96	92.7	40L /96	93.8
84R /573	84074-85792	Putative ATPase-dependent protease	84R /573	100.0	86R /573	98.8	86R /573	98.8	86R /573	98.8	79R /573	98.6	28L /573	95.2	26L /577	86.5
No. of total ORFs identified			104		106		104		105		99		102		96	
Overall similarity (%) of total 104 ORFs			99.8		91.5		92.2		84.2		86.2		83.1		69.6	
No. of not applicable/<90% homology ORFs out of total 104 ORFs (%)			0/0		6/5 (10.6)		6/4 (9.6)		15/2 (16.3)		12/4 (15.4)		13/7 (19.2)		25/9 (32.7)	
No. of not applicable/<90% homology ORFs out of 10 functional ORFs (%)			0/0		0/1		1/2		2/2		0/3		1/2		3/6	

## Data Availability

All sequence data used in this study are available in Appendix A or in the GenBank.

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
