# Peer review of "First Report of Endemic Frog Virus 3 (FV3)-like Ranaviruses in the Korean Clawed Salamander (*Onychodactylus koreanus*) in Asia"

_viruses, 2024, doi:10.3390/v16050675_

Round 1
Reviewer 1 Report
Comments and Suggestions for Authors
The report addresses the question whether or not FV3-like viruses found in Asia resulted from importation from North America or another region, or whether FV3-like viruses are endemic to Asia. The authors found that an FV3-like virus infected salamanders present in a high mountain stream and suggest an endemic origin of these viruses. Furthermore phylogenetic analysis suggests that the Asian isolates comprise a clade distinct from those viruses found in North America. Specific comments follow.
1. Line 13 and elsewhere: Iridoviridae must be in italics when referring to the family.
2. Line 39: Unless you are referring to the genus Ranavirus, the word ranavirus is not italicized. For example, “Ranaviruses infect frogs, fish and reptiles.” Furthermore, consider rewriting this sentence to indicate that three ranavirus species infect amphibians, whereas three other ranavirus species infect primarily fish.
3. Line 48: Change to “spotted salamander Maine virus, SSME; frog virus 3 isolate RUK13, RUK13; …” Also be consistent in the use of commas (,) and semi-colons (;).
4. Line 69: Delete “cultured the cells” since I do not see evidence of establishing cell lines from these animals.
5. Line 79: What is “Chuncheon-si?” If it is a place, province, etc. please indicate who is granting permission.
6. Line 125 and elsewhere: Since Table S1 links FV3 with the original isolate, perhaps the term you should use here (and elsewhere) is “FV3-like” viruses. Since the boundaries between species and strains are unclear, most all viruses you are referring to (in Fig. 3 and Table S2) are isolates of the species Frog virus 3 (hence FV3-like viruses) with the exception of CMTV and ATV which are distinct species according to the ICTV.
7. Line 145: What are the 47 core genes used to perform the phylogenetic analysis? Among all genera within the family, generally 23 – 26 core genes are recognized. I suspect if only ranaviruses are considered the number is higher, but what are the 47. A table or an indication in Table S2, e.g., by displaying the ORF in boldface type, would be appropriate. The reference given in support of 47 core genes (#14) is to a PhD thesis that may not be readily available.
8. Line 177: Change to “…(encoding 40 functional and 64 hypothetical proteins) ranging in size from 43 to 1294 amino acids in length.”
9. Line 181ff: When first reading this paragraph I came away (based on Table 1) with the idea that ATV only shared 69.7% similarity with OKRV1, but when examining Table S2 it was clear that many ATV ORFs shared high similarity with OKRV1 ORFs. What do the authors think? Is ATV really that much different than OKRV1 or do a number of hypothetical ORFs skew the analysis? Are these hypothetical ORFs part of he 47 core genes? Please clarify.
10. Fig 3: OKRV2 is part of the Asian clade along with STIV. STIV and FV3 display a truncated vIF-2 protein. How common is that truncation among both Asian and North American isolates?
11. Table 1: This table is very difficult to read since the data in each column is disordered/jumbled. Perhaps it could be simplified by dropping one or two of the viruses to improve readability. At the bottom of the table, how was the value 69.57% of ATV ORFs similar to those of OKRV1 arrived at?
12. Line 258: Consider changing to: “STIV was detected in farmed reptiles and soft-shelled turtles in Shenzen, a coastal city in China [50]. Therefore, the origin of STIV…”
13. Line 265: Consider changing to “…and Triturus) that can be infected by FV3-like viruses [51-56].” The change is advised because amphibians do not infect FV3 and the term worldwide implies NA and Europe, but also likely includes Asia, Africa, and South America.
14: Overall assessment: The question of whether FV3-like viruses are endemic to Asia or introduced from another location remains unanswered. The tree shown in Fig. 3 supports the view that Asian isolates occupy a clade apart from those from the New World, but the presence of an introduced strain (STIV) within the Asian clade along with presumably endemic strains (OKRV1 and OKRV2) leaves the question of whether Asia strains are endemic or introduced unanswered. Another question/aspect is the existence of the truncated vIF-2 alpha ORF in FV3 and STIV. How common is this truncation? Can it be used to track viral lineages? Clearly, the studies provided here help us understand the origin of FV3-like viruses, but do not provide a definitive answer and the authors end their work stating that additional studies need to be done.
Comments on the Quality of English LanguageSee above for suggested grammatical changes.
Author Response
Dear Reviewer 1,
We appreciate your valuable comments and the time spent on our manuscript. The comments significantly improved our work. We have considered all of your comments, provided our answers to each of them, and revised the manuscript accordingly. All our edits can be viewed with “track changes”. Our responses to the reviewers’ comments are written in blue text in the attached file.
Sincerely yours,
Daesik Park

Reviewer 2 Report
Comments and Suggestions for Authors
The manuscript presents the isolation, genomic sequencing and phylogenic analysis of two FV3-like ranviruses from a local Urodele Salamander. The study design is straightforward and results add our understanding to ranavirus endemic in the wild amphibians in Asia. Some major points are listed below for revision:
The title could be restructured as “First Report of the Endemic Frog Virus 3 (FV3)-like Ranavirus in the Korean Clawed Salamander (Onychodactylus koreanus) in Asia”
Line 41-43: FV3-like Ranavirus seemed to be used interchangeably with FV3 virus. Make this consistent please.
Line 47, 53, 55: I think they meant to use Ranavirus instead of FV3 viruses. The listed viruses are types of Ranavirus. This should be crosschecked across the writeup too.
Line 141-142: Phylogenomic or phylogenetic? I suggest to use phylogenetic unless the analysis was conducted at a whole genome level.
Line 177: 43 to 1294 AA long?
Table 1 seems to be overpacked, restructuring for more clarity is encouraged or it should be a supplemental table using an Excel sheet, for example.
Line 285: ...among the viruses that were previously found in China.
286: (RNRV) (insert comma instead of and) North America and (delete “also”) ….
Comments on the Quality of English LanguageSee that above
Author Response
Dear Reviewer 2,
We appreciate your valuable comments and the time spent on our manuscript. The comments significantly improved our work. We have considered all of your comments, provided our answers to each of them, and revised the manuscript accordingly. All our edits can be viewed with “track changes”. Our responses to the reviewers’ comments are written in blue text in the attached file.
Sincerely yours,
Daesik Park
